



# Novel hydrocarbon-utilizing soil mycobacteria synthesize unique mycocerosic acids at a Sicilian everlasting fire

Nadine T. Smit[1], Laura Villlanueva[1], Darci Rush[1], Fausto Grassa[2], Caitlyn R. Witkowski[1], Mira Holzheimer[3], Adriaan J. Minnaard[3], Jaap S. Sinninghe Damsté[1,4], Stefan Schouten[1,4]

[1] NIOZ Royal Netherlands Institute for Sea Research, Department of Marine Microbiology and Biogeochemistry, and Utrecht University, P.O. Box 59, 1790 Ab Den Burg Texel, the Netherlands

[2] Istituto Nazionale di Geofisica e Vulcanologia, Sezione di Palermo, Via Ugo La Malfa, 153, 90144, Palermo, Italy

[3] Stratingh Institute for Chemistry, University of Groningen, 9747 AG Groningen, the Netherlands

[4] Department of Earth Sciences, Faculty of Geosciences, Utrecht University, P.O. Box 80.121, 3508 TA Utrecht, the Netherlands

*Correspondence to* Nadine T. Smit (nadine.smit@nioz.nl)

**Abstract.** Soil bacteria rank among the most diverse groups of organisms on Earth and actively impact global processes of carbon cycling, especially in the emission of greenhouse gases like methane, $CO_2$ and higher gaseous hydrocarbons. An abundant group of soil bacteria are the mycobacteria, which colonize various habitats due to their impermeable cell envelope that contains remarkable lipids. These bacteria have been found to be highly abundant at petroleum and gas seep areas, where they might utilize the released hydrocarbons. However, the function and the lipid biomarker inventory of these soil mycobacteria are poorly studied. Here, soils from the Fuoco di Censo seep, an everlasting fire (gas seep) in Sicily, Italy, were investigated for the presence of mycobacteria via 16S rRNA gene sequencing and fatty acid profiling. The soils contained high relative abundances (up to 34 % of reads assigned) of mycobacteria, phylogenetically close to the *Mycobacterium simiae* complex and more distant to the well-studied *M. tuberculosis* and hydrocarbon-utilizing *M. paraffinicum*. The soils showed decreasing abundances of mycocerosic acids (MAs), fatty acids unique for mycobacteria, with increasing distance from the seep. The major MAs at this seep were tentatively identified as 2,4,6,8-tetramethyl tetracosanoic acid and 2,4,6,8,10-pentamethyl hexacosanoic acid. Unusual MAs with mid-chain methyl branches at positions C-12 and C-16 (i.e. 2,12-dimethyl eicosanoic acid and 2,4,6,8,16-pentamethyl tetracosanoic acid) were also present. The molecular structures of the Censo MAs are different from those of the well-studied mycobacteria like *M. tuberculosis* or *M. bovis* and have relatively [13]C-depleted values (-38 to -48 ‰), suggesting a direct or indirect utilization of the released seep gases like methane or ethane. The structurally unique MAs in combination with their depleted $\delta^{13}C$ values identified at the Censo seep offer a new tool to study the role of soil mycobacteria as hydrocarbon gas consumers in the carbon cycle.

30



# 1. Introduction

Soils harbor the largest diversity of microorganisms on our planet and have a large influence on the Earth's ecosystem as they actively impact nutrient and carbon cycling, plant production and the emissions of greenhouse gases (Tiedje et al., 1999; Bardgett and van der Putten, 2014; Delgado-Baquerizo et al., 2018). Soil bacteria rank among the most diverse and abundant groups of organisms on Earth. However, numerous studies suggest that most of their function and diversity in our ecosystems is still undescribed (Tiedje et al., 1999; Bardgett and Van Der Putten, 2014). The assessment of soil bacterial diversity has mainly relied on 16S ribosomal RNA (rRNA) gene sequencing and has indicated that the most abundant bacterial phylotypes in global soils include Alphaproteobacteria, Gammaproteobacteria, Betaproteobacteria, Actinobacteria, Acidobacteria and Planctomycetes (Fierer et al., 2012; Delgado-Baquerizo et al., 2018). Besides the use of DNA-based techniques, lipid biomarkers offer an additional tool to investigate soil bacterial communities, such as fatty acids specific for methanotrophs (Bull et al., 2000; Bodelier et al., 2009) or branched glycerol dialky glycerol tetraether (brGDGTs) believed to derive from soil acidobacteria (Weijers et al., 2009; Peterse et al., 2010; Sinninghe Damsté et al., 2018).

Mycobacteria of the genus *Mycobacterium* belonging to the phylum Actinobacteria form an abundant microbial group in global soils (Falkinham, 2015; Walsh et al., 2019). Some members of the genus *Mycobacterium* are obligate pathogens (e.g. *Mycobacterium tuberculosis* and *Mycobacterium leprae*) and are the cause of more than 1.5 million annual human deaths worldwide through the diseases tuberculosis and leprosy (World Health Organization, 2019) and have consequently been more frequently studied than opportunistic pathogenic and non-pathogenic environmental mycobacteria. Interestingly, early studies from the 1950s reported high abundances of non-pathogenic hydrocarbon-consuming mycobacteria (*M. paraffinicum*) in areas of oil and gas production, gas seeps, and in common garden soils (Davis et al., 1956; Dworkin and Foster, 1958; Davis et al., 1959). Cultivation and genomic studies show that mycobacteria can oxidize a range of greenhouse gases (ethane, propane, alkenes, carbon monoxide or hydrogen) and can degrade toxic polycyclic aromatic hydrocarbons (Miller et al., 2007; Hennessee et al., 2009; Coleman et al., 2012; Martin et al., 2014). Mycobacteria are able to colonize a wide variety of habitats from soils to aquatic and human-engineered environments (Brennan and Nikaido, 1995; Falkinham, 2009). Their impermeable cell envelope may play an important role in their ecological dominance. It consists of a peptidoglycan polymer that is surrounded by a thick hydrophobic lipid-rich outer membrane. This impermeability favors the formation of biofilms, thus enabling mycobacteria to be often the first colonizers at environmental interfaces like air-water or surface-water (Brennan and Nikaido, 1995). Additionally, the impermeable cell membrane allows a resistance to acidic conditions, anoxic survival and the possibility to metabolize recalcitrant carbon compounds. Mycobacteria feature an unusual lipid inventory such as extremely long fatty acids with chains up to 90 carbon atoms long and with numerous methyl groups, hydroxylations and/or methoxylations (Minnikin et al., 1985; Minnikin et al., 1993a; Donoghue et al., 2017; Daffé et al., 2019). Furthermore, mycobacteria are known to synthesize fatty acids like



tuberculostearic acid (10-Me $C_{18:0}$) and the multi methyl-branched mycocerosic acids (MAs) that can contain 3 to 5 methyl branches at regularly spaced intervals such as at positions C-2, C-4, C-6 and C-8 (Minnikin et al., 1985; Minnikin et al., 1993a; Minnikin et al., 2002; Redman et al., 2009). These unusual fatty acids are bound to complex glycolipids like phthiocerol dimycocerosates (PDIMs), diacyl trehalose (DATs) or phenolic glycolipids (Minnikin et al., 2002; Jackson et al., 2007). However, in contrast to the pathogenic and opportunistic pathogenic mycobacteria, the lipid biomarker inventory of non-pathogenic mycobacteria in soils and other environments remains poorly described.

In this study, we investigated soils near a continuous gas seep named "Fuoco di Censo" ("Everlasting Fire") in Sicily, Italy to explore the presence of non-pathogenic, potentially hydrocarbon-utilizing, mycobacterial species using 16S rRNA gene amplicon sequencing and fatty acid profiling. It resulted in the identification of potential biomarkers for the presence of mycobacteria in terrestrial environments and hydrocarbon seeps. Furthermore, their stable carbon isotopic composition provided hints for their role in the carbon cycle in this gas seepage environment.

## 2. Material and Methods

### 2.2 Study area

The Fuoco di Censo seep (Censo seep; 39º62.503'N, 13º38.751'E) is located at 800 m above sea level in the mountains of Southwestern Sicily, Italy (Etiope et al., 2002; Grassa et al., 2004). The area is part of the Alpine orogenic belt in the Mediterranean and located along the boundary of the African and European plates (Basilone, 2012). The Censo seep belongs to the Bivona area, which is characterized by a complex geological setting. The seep is located in an area with sandy clays, marls and evaporites from the Tortonian-Messinian that are covered by a thrusting limestone of Carnian-Rhetian age (Trincianti et al., 2015). The Censo seep is a typical example of a natural 'Everlasting Fire', which is characterized by the absence of water and the temporal production of flames, that can be several meters high, by a continuous gas flux (Etiope et al., 2002). The Censo seep gas consists mainly of $CH_4$ (76-86 %) and $N_2$ (10-17%) as well as some other minor gases like $CO_2$, $O_2$, ethane, propane, He and $H_2$ (Etiope et al., 2002; Grassa et al., 2004). A diffuse soil degassing is detectable within an area of 80 $m^2$ with an average $CH_4$ flux of $7*10^6$ mg $m^{-2}$ $d^{-1}$ and a total $CH_4$ emission of $6.2*10^3$ kg yr $^{-1}$ (Etiope et al., 2002; Etiope et al., 2007). The $CH_4$ is suggested to be generated by the thermal alteration of organic matter and is characterized by a stable carbon isotopic composition of $\delta^{13}C$ = -35 ‰ and $\delta^2H$ = -146 ‰ (Grassa et al., 2004). This thermogenic $CH_4$ possibly derives from mature marine source rocks (kerogen type II) with a thermal maturity beyond the oil window, resulting in a dry gas with $C_1/(C_2+C_3)$ ratios greater than 100 (Grassa et al., 2004).

### 2.2 Sample collection

Soil samples of the Censo seep were recovered during a field campaign in October 2017. The soil was collected from a horizon 5 to 10 cm below the surface and at three distances from the seep, i.e. 0 m (seep site), 0.8 m, and a control at 13.2 m



distance from the main vent. The *in-situ* temperature of the soils at the time of collection was ca. 18 ºC. The soils were directly transferred into a clean geochemical sampling bag and stored frozen at -20 ºC until freeze drying and extraction.

## 2.3 Extraction and saponification

Freeze dried Censo soils were extracted with a modified Bligh and Dyer extraction (Schouten et al., 2008; Bale et al., 2013).
Soil samples (ca. 12 g) were ultrasonically extracted (10 min) with a solvent mixture containing methanol (MeOH), dichloromethane (DCM) and phosphate buffer (2: 1: 0.8, v: v: v). After centrifugation, the solvent was collected, combined and the residues re-extracted twice. A biphasic separation was achieved by adding additional DCM and phosphate buffer to a ratio of MeOH, DCM and phosphate buffer (1: 1: 0.9, v: v: v). The aqueous layer was washed two more times with DCM and the combined organic layers dried over a $Na_2SO_4$ column followed by drying under $N_2$.
Saponification (base hydrolysis) was conducted on aliquots (1-7 mg) of the Bligh Dyer extracts (BDEs) by the addition of 2 ml 1N KOH in MeOH solution and refluxing for 1 h at 130 °C. After cooling, the pH was adjusted to 5 by using a 2N HCL in MeOH solution, separated with 2 ml bidistilled water and 2 ml DCM, and the organic bottom layer was collected. The aqueous layer was washed two more times with DCM and the combined organic layers dried over a $Na_2SO_4$ column followed by drying under $N_2$.


## 2.4 Derivatization of fatty acids

### 2.4.1 Preparation of fatty acid methyl esters using BF₃

Aliquots of the saponified Censo seep BDEs and aliquots of a mycocerosic acid standard (2,4,6-trimethyl-tetracosanoic acid; $C_{27}$ MA standard) synthesized by hydrogenation with palladium and charcoal from mycolipenic acid (Holzheimer et al.,
2020), were esterified with 0.5 ml of a boron trifluoride-methanol solution ($BF_3$ solution) for 10 min at 60 °C. After cooling, 0.5 ml bidistilled water and 0.5 ml DCM were added and shaken, and the DCM bottom layer pipetted off. The water layer was extracted twice with DCM and the combined DCM layers were dried over an $MgSO_4$ column. The soil extracts were eluted over a small silica gel column with ethyl acetate as an eluent to remove polar material. Extracts were subsequently separated using a small column packed with activated aluminum oxide into two fractions. The first fraction (fatty acid
methyl ester fraction) was eluted with 4 column volumes of DCM followed by a second fraction (polar fraction) eluted with 3 column volumes of DCM/MeOH (1:1). The fatty acid methyl ester fractions were dried under a continuous flow of $N_2$ and analyzed using gas chromatography-mass spectrometry (GC-MS) and GC-isotope ratio mass spectrometry (IRMS).

### 2.4.2 Preparation of fatty acid "picolinyl esters" derivatives using 3-pyridylcarbinol



Aliquots of saponified Censo seep BDEs, as well as aliquots of the $C_{27}$ MA standard, were derivatized into picolinyl esters. This technique enhances the abundance of diagnostic fragment ions in the mass spectrum, such as those of methyl branching points in fatty acids, enabling an improved structural identification (Christie, 1998; Harvey, 1998). Different 'picolinyl' derivatization protocols were tested on the $C_{27}$ MA standard and the highest yields were achieved by the procedure in Harvey, 1998. In this procedure, 0.5 ml of thionyl chloride was added using a 1 ml disposable syringe to 1 mg aliquot of the

dried saponified Censo seep BDEs in a pressure vial and left for ca. 2 min at room temperature. The vials were then dried by a continuous flow of $N_2$. 0.5 ml of a 1% 3-pyridylcarbinol in acetonitrile solution was added in the reaction vials and left at room temperature for 2 min. The volumes of reagents in this protocol were reduced (0.1 ml) for 0.1 mg of the MA standard. The 'picolinyl' esters were transferred with acetonitrile to 2 ml analysis vials and the concentration was adjusted to 1 mg/ml with acetonitrile. The 'picolinyl' esters were analyzed using GC-MS with acetonitrile as injection solvent.


### 2.4.3 Preparation of fatty acid methyl sulfide esters using dimethyl disulfide (DMDS)

To determine the position of the double bonds in unsaturated fatty acids, dimethyl disulfide (DMDS) derivatization was used (Francis, 1981; Nichols et al., 1986). For this, 100 µl of hexane, 100 µl of DMDS solution (Merck ≥ 99%) and 20 µl of $I_2$/ether were added to the dry aliquot and heated overnight at 40 ºC. The mixture was left to room temperature and 400 µl of

hexane and 200 µl of a 5% aqueous solution of $Na_2S_2O_3$ (for iodine deactivation) were added and mixed. The upper hexane layer was removed, and the aqueous layer washed twice with hexane. The three hexane layers were combined and dried over a $Na_2SO_4$ column before GC-MS analysis with hexane as injection solvent.

### 2.5 Instrumental analysis

### 2.5.1 Gas chromatography-mass spectrometry (GC-MS)

GC-MS was performed using an Agilent Technologies GC-MS Triple Quad 7000C in full scan mode. A CP-Sil5 CB column (25 m x 0.32 mm with a film of 0.12 µm, Agilent Technologies) was used for the chromatography with He as carrier gas (constant flow 2 ml min$^{-1}$). The samples (1 µl) were injected on column at 70 °C, the temperature was increased at 20 °C min$^{-1}$ to 130 °C, raised further by 4 °C min$^{-1}$ to 320 °C, at which it was held for 20 min. The mass spectrometer was operated

over a mass range of *m/z* 50 to 850, the gain was set on 3, with a scan time of 700 ms.

### 2.5.2 Gas chromatography-isotope ratio mass spectrometry (GC-IRMS)

GC-IRMS was carried out with a Thermo Scientific Trace 1310 with a GC-Isolink II, a ConFlo IV and a Delta Advantage IRMS. The gas chromatography was performed on a CP-Sil5 CB column (25 m x 0.32 mm with a film thickness of 0.12 µm,

Agilent) with He as carrier gas (constant flow 2 ml min$^{-1}$). The $BF_3$ methylated samples (dissolved in ethyl acetate) were on-



column injected at 70 ºC and subsequently, the oven was programmed to 130 ºC at 20 ºC min$^{-1}$, and then at 4 ºC min$^{-1}$ to 320 ºC, which was held for 10 min. Stable carbon isotope ratios are reported in delta-notation against VPDB $^{13}$C standard. Values were determined by two analysis and results averaged to a mean value.

**2.6 DNA extraction, 16S rRNA gene amplification, analysis and phylogeny**

DNA was extracted from sediments using the PowerMax soil DNA isolation kit (Qiagen). DNA extracts were stored at -80 °C until further analysis. The 16S rRNA gene amplicon sequencing and analysis was performed with the general 16S rRNA archaeal and bacterial primer pair 515F and 806RB targeting the V4 region (Caporaso et al., 2012; Besseling et al., 2018). Polymerase chain reaction (PCR) products were gel purified using the QIAquick Gel-Purification kit (Qiagen), pooled and

diluted. Sequencing was performed at the Utrecht Sequencing Facility (Utrecht, the Netherlands) using an Illumina MiSeq sequencing platform. The 16S rRNA gene amplicon sequences were analyzed by an in-house pipeline (Asbun et al., 2019) that includes quality assessment by FastQC (Andrews, 2010), assembly of the paired-end reads with Pear (Zhang et al., 2013), and assignment of taxonomy (including picking representative set of sequences with 'longest' method) with blast by using the Silva 128 release as a reference database (https://www.arb-silva.de/). Representative operational taxonomic unit

(OTU) sequences (assigned with OTU picking method based on 97 % nucleotide similarity with Uclust) (Edgar, 2010), attributed to the family Mycobacteriaceae were aligned by using Muscle (Edgar, 2004) implemented in MEGA6, and then used to construct a phylogeny together with 16S rRNA gene sequences of characterized *Mycobacterium* species and closely related uncultured Mycobacteriaceae 16S rRNA gene sequences. The phylogenetic tree was inferred using the Maximum Likelihood method based on the General Time Reversible model (Nei and Kumar, 2000). The analysis involved 32

nucleotide sequences with 294 base pairs positions in the final dataset. Evolutionary analyses were conducted in MEGA6 (Tamura et al., 2013).

## 3. Results and discussion

### 3.1 Microbial diversity in the Censo seep soils

Soils were sampled at the Censo seep and with increasing distance from the seep (Table 1). To investigate the microbial diversity, 16S rRNA gene libraries were generated from extracted DNA using 16S rRNA gene amplicon sequencing. This analysis showed a high relative abundance of 16S rRNA gene reads attributed to Mycobacteriaceae ranging from 0.7 to 34.1 % of assigned bacterial plus archaeal reads in the soils with relative abundances increasing with decreasing distance to the seep (Table 1). Sequences assigned to known methanotrophs are Gammaproteobacteria (Methylococcales),

Alphaproteobacteria (Methylocystaceae and Methylobacteriaceae) and Verrucomicrobia ('*Candidatus* Methylacidiphilum') but only accounted for 0.2 to 5.1 % of the total number of reads assigned (Table 1). Phylogenetic analysis indicated that there are two sequences representative for operational taxonomic units (OTUs) attributed to mycobacteria (i.e. sequences





Censo seep 1 and Censo seep 2) present in the soils (Fig. 1). Both OTUs are phylogenetically most closely related to sequences of the *Mycobacterium simiae* complex (Tortoli, 2014) (Fig. 2; >98 % identical considering the 294 bp sequence

fragment analyzed), which include *M. simiae*, *M. europaeum*, *M. kubicae* and *M. heidelbergense* (Hamieh et al., 2018). Previously described cultivated mycobacteria of the *M. simiae* complex are slow-growing mycobacterium species isolated from environmental niches but also associated to infections in humans as opportunistic pathogens (Lévy-Frébault et al., 1987; Heap, 1989; Bouam et al., 2018). The Censo seep sequences are more distantly related (94-95 % identical) to frequently studied pathogenic mycobacteria (such as *M. tuberculosis* and *M. leprae*) and other environmental mycobacteria

like hydrocarbon-utilizers (e.g. *M. paraffinicum* and *M. vanbaalenii*) (Fig. 1). To the best of our knowledge the hydrocarbon-utilizing bacteria have not been isolated from humans or animals (e.g. *M. vanbaalenii*) and are mostly able to degrade aromatic hydrocarbons (Kweon et al., 2015). These data reveal the presence and relevance of uncultured mycobacteria in the soils around the Censo seep. This is in line with previous reports of the occurrence of mycobacteria near petroleum seeps and gas fields (Davis et al., 1956; Davis et al., 1959).


**3.2 Fatty acid composition of Censo seep soils**

Analysis of the fatty acid fractions of the Censo seep soils reveal a distinct pattern that changes with increasing distance from the main seep (Fig. 2). Common fatty acids such as $C_{16:0}$, $C_{16:1\omega6}$, $C_{16:1\omega7}$, $C_{18}$, $C_{18:1\omega9}$ and $C_{18:1\omega7}$ as well as the longer chain $C_{22}$ and $C_{24}$ fatty acids occur in all three soils. $C_{16}$ and $C_{18}$ fatty acids are abundant lipids in soils and are synthesized by

diverse bacteria and fungi, whereas the longer chain ($C_{22}$-$C_{24}$) fatty acids originate commonly from higher plants (Řezanka and Sigler, 2009; Frostegård et al., 2011). These fatty acids could also derive from mycobacteria which can produce fatty acids ($C_{14}$ to $C_{26}$) with high amounts of $C_{16}$ and $C_{18}$ fatty acids and their unsaturated homologues (Chou et al., 1996; Torkko et al., 2003). Besides mycobacteria which are abundant in the soils close to the main seepage (Table 1), the $C_{16}$ fatty acids may also originate from Type I methanotrophs (Gammaproteobacteria), whereas $C_{18}$ fatty acids could derive from type II

methanotrophs (Alphaproteobacteria), present in these Censo seep soils (Fig. 2 and Table 1) (Bull et al., 2000; Bowman et al., 1993; Bodelier et al., 2009). Although the relative abundances of 16S rRNA gene reads of this Type I and II methanotrophs are only minor in the Censo soils (Table 1).

The Censo seep soils also feature $C_{31}$-$C_{33}$ 17β,21β(H)-homohopanoic acids, the most abundant of which is the $C_{32}$ 17β,21β(H)-hopanoic acid (bishomohopanoic acid) (Fig. 2). Hopanoic acids are common components in terrestrial

environments (Ourisson et al., 1979; Rohmer et al., 1984; Ries-Kautt and Albrecht, 1989 ;Crossman et al., 2005) and can be derived from a range of bacteria, including Alpha-and Gammaproteobacteria, Planctomycetes and Acidobacteria (Thiel et al., 2003; Sinninghe Damsté et al., 2004; Birgel and Peckmann, 2008; Sinninghe Damsté et al., 2017). Explorative searches of genomic databases for the biosynthetic gene encoding squalene-hopane-cyclase (shc) in mycobacteria from the *M. simiae* complex revealed a potential for biohopanoid production. Therefore, mycobacteria may be an additional source for hopanoic

acids in the Censo seep soils.



Interestingly, at the seep (0 m) the FA pattern is dominated by unusual FAs ranging from $C_{19}$ to $C_{31}$, which are absent further away from the main seepage (Fig. 2). The mass spectra of the three most abundant representatives of these fatty acids are shown in Figure 3. Mass spectra of the methyl ester derivatives of these fatty acids show major fragment ions of $m/z$ 88 and 101. These fragments result from "McLafferty" rearrangements associated with the presence of the carboxylic

acid methyl ester group (Lough, 1975; Ran-Ressler et al., 2012). The presence of the even-numbered $m/z$ 88 fragment ion, rather than the typical fragment ion at $m/z$ 74 in the mass spectra of methyl esters of $n$-FAs, strongly suggests a methyl at position C-2 (Fig. 3). One FA shows also high fragment ions at $m/z$ 213 and $m/z$ 241 (Fig. 3A). This difference of 28 Da hints at a second methyl group at position C-12 (Fig. 3A). Two of these fatty acids show a fragment ion at $m/z$ 129 (Figs. 3C and E), suggesting the presence of an additional methyl at position C-4 of the fatty acids. The apparent methyl branches in

these fatty acids are in agreement with the relatively early retention times of these FAs compared to the regular straight-chain counterparts (Fig. 2). Other fragment ions, including those potentially revealing the positions of additional methyl groups, were only present in low abundance, complicating further structural identification. Nevertheless, the presence of methyl branches at C-2 and C-4 in a number of these fatty acids does suggest that they may be related to mycobacteria-derived MAs, which share the same structural characteristics (Alugupalli et al., 1998; Nicoara et al., 2013). Indeed, the mass

spectrum of the methyl ester of a synthetic $C_{27}$ MA standard (2,4,6-trimethyl-tetracosanoic acid) (Holzheimer et al., 2020) shows identical mass spectral features (i.e. m/z 88 and 129; Fig. 4A). However, full structural interpretation of the mass spectrum of this authentic standard is also complicated by the low abundances of diagnostic fragment ions indicative for the position of the methyl branches in the alkyl chain.

To enhance the diagnostic fragmentation patterns of these potential MAs, the fatty acids were also transformed into

a "picolinyl ester" (Harvey, 1998). The potential of this technique is revealed by the mass spectrum of the synthetic MA standard (2,4,6-trimethyl-tetracosanoic acid) "picolinyl ester" derivative (Fig. 4B), which shows fragment ions revealing all positions of methylation of the fatty acid $n$-alkyl chain. The high intensity of the fragment ion of $m/z$ 165 indicates the presence of a methyl group at position C-2, while the presence of the fragments ions at $m/z$ 178 and 206 combined with the absence of an $m/z$ 192 fragment ion indicates the presence of a methyl at C-4. Similarly, the presence of the third methyl at

position C-6 is revealed by the fragment ions at $m/z$ 220 and 248 and the low abundance of the fragment ion at $m/z$ 234. Thus, the "picolinyl derivatization" technique substantially increases the confidence in the structural identification of multi-methyl-branched fatty acids using mass spectrometry. Therefore, this "picolinyl ester" derivatization technique was also applied to determine the methylation pattern of the potentially novel MAs in the Censo seep soils (Fig. 3).

To illustrate this approach, we discuss the identification of the three major MAs. When analyzed as "picolinyl ester

derivatives" (Fig. 3B, D and F), these MAs respectively showed molecular ions (M+) of $m/z$ 431, 515, and 557, indicating $C_{22}$, $C_{28}$, and $C_{31}$ MAs. The mass spectrum of the "picolinyl ester derivative" $C_{28}$ MA (Fig. 3D), the most abundant MA in the Censo seep soils, confirms the methylation at C-2 with the fragment ion of $m/z$ 165. Furthermore, this spectrum also shows abundant fragment ions at $m/z$ 178, 206, 220, 248, 262 and 290. Combined with the absence of the fragment ions at $m/z$ 192, 234 and 276, this strongly suggests the presence of three additional methyl groups at position C-4, C-6 and C-8.





The mass spectrum of the $C_{22}$ "picolinyl ester derivative" (Fig. 3B) also confirms the methyl branch at position C-2 through the mass ion $m/z$ 165. Elevated fragment ions at $m/z$ 290 and 318 in combination with the low intensity of the fragment ion at $m/z$ 304 suggests a methyl group at position C-12. Further mass spectral interpretations can be made for the $C_{31}$ MA, with mass spectrum similar to that of the $C_{28}$ MA but including an additional methyl group at position C-10, as indicated by the presence of fragment ions at $m/z$ 304 and 332 and the absence of a fragment ion at $m/z$ 318 (Fig. 3F). Thus, we tentatively

identified these MAs as 2,12-dimethyl-eicosanoic acid ($C_{22}$ 2,12-dimethyl MA), 2,4,6,8-tetramethyl-tetracosanoic acid ($C_{28}$ 2,4,6,8-tetramethyl MA) and 2,4,6,8,10-pentamethyl-hexacosanoic acid ($C_{31}$ 2,4,6,8,10-pentamethyl MA), respectively (Figs. 2, 3 and Table 2). Other abundant MAs tentatively identified include 2-methyl-octadecanoic acid ($C_{19}$ 2-methyl MA), 2-methyl-nonadecanoic acid ($C_{20}$ 2-methyl MA), 2-methyl-eicosanoic($C_{21}$ 2-methyl MA), 2,4,6-trimethyl-docosanoic acid ($C_{25}$ 2,4,6-trimethyl MA), 2,4,6,8-tetramethyl-pentacosanoic acid ($C_{29}$ 2,4,6,8-tetramethyl) and 2,4,6,8,16-pentamethyl-

tetracosanoic acid ($C_{29}$ 2,4,6,8,16-pentamethyl MA) MAs (Fig. 2 and Table 2).

At the seep (0 m), the MAs have a high relative abundance, representing ca. 44% of the total FAs. Their abundance decreases to ca. 20% in the soil at 1.8 m from the seep, whereas MAs were not detected in the soil at 13.2 m distance from the seep (Fig. 2). These lipids show a similar distribution trend as the 16S rRNA gene sequencing results, which show high relative abundances of sequences from mycobacteria at the seep (ca. 34.1 % at 0 m), decrease to 8.5 % at 1.8 m, and are <1

% at 13.2 m (Table 1). Therefore, both the specific structure and the 16S rRNA gene data strongly suggest that the unusual FAs are derived from mycobacteria.

### 3.3 Mycocerosic acids as biomarkers for mycobacteria in the environment

Mycocerosic acids have been mainly studied as biomarker for diseases from pathogenic (e.g. *M. tuberculosis* or *M. leprae*)

or opportunistic pathogenic (e.g. the *M. simiae* complex) mycobacteria in the last decades (e.g. Minnikin et al., 1993a; Minnikin et al., 1993b; Torkko et al., 2003). These studies revealed a high structural variability of MAs with distribution patterns characteristic for different mycobacterial species. For example, the frequently studied *M. tuberculosis* shows a major $C_{32}$ 2,4,6,8-tetramethyl MA and *M. leprae* a $C_{34}$ 2,4,6,8-tetramethyl MA (Minnikin et al., 1993a; Minnikin et al., 1993b), whereas other mycobacteria feature shorter chain major MAs like $C_{21}$ 2-methyl MA in *M. palustre* and $C_{22}$ 2,4-dimethyl MA

in *M. intermedium* (Chou et al., 1996; Torkko et al., 2002) (Fig. 2 and Table 2).

The Censo seep soils reveal a high number of tentatively identified MAs which have not been reported previously (Fig. 1 and Table 2), e.g. those biosynthesized by pathogenic mycobacteria like *M. tuberculosis* and *M. leprae* and by mycobacteria belonging to the more closely related *M. simiae* complex like *M. heidelbergense* (Minnikin et al., 1993a; Minnikin et al., 1993b; Torkko et al., 2003). The MA distribution of the Censo seep soils is characterized by a dominant $C_{28}$

2,4,6,8-tetramethyl MA, while the MA distribution of *M. heidelbergense* or *M. palustre* from the *M. simiae* complex is dominated by the $C_{21}$ 2-methyl MA. Other more distantly related environmental opportunistic pathogens besides those of the *M. simiae* complex, like *M. marinum* or *M. intermedium,* produce a dominant 2,4,6- $C_{27}$ trimethyl or $C_{22}$ 2,4-dimethyl MA.





As mentioned earlier, pathogenic mycobacteria like *M. tuberculosis* feature a major $C_{32}$ 2,4,6,8-tetramethyl MA as well as *M. leprae* produces a dominant $C_{34}$ 2,4,6,8-tetramethyl MA, clearly different from the major MA in the Censo soils (Fig. 1 and
Table 2). Possibly, these unusual MAs could help to differentiate environmental Censo mycobacteria from opportunistic pathogenic and pathogenic mycobacteria in various modern and past environments.

Interestingly, the Censo mycobacteria show relatively high abundances of pentamethylated MAs ($C_{29}$ 2,4,6,8,16-pentamethyl MA and $C_{31}$ 2,4,6,8,10-pentamethyl MA) compared to other studied mycobacteria. *M. kansasii* has a dominant pentamethyl MA ($C_{33}$ 2,4,6,8,10-pentamethyl MA; Table 2), which was also been reported in *M. tuberculosis* and *M. leprae*
albeit in very low abundances, while *M. botniense* features a partially identified pentamethylated $C_{27}$ (2,4,6,x,x) MA (Minnikin et al., 1985; Daffé and Laneelle, 1988; Torkko et al., 2003). Shorter chain MAs are also abundant in the Censo soils, some of which have been identified in other mycobacterial species (Fig. 1 and Table 2): $C_{20}$ 2-methyl MA (*M. palustre*), $C_{21}$ 2-methyl MA (e.g. *M. palustre*, *M. heidelbergense* or *M. interjectum*) and $C_{25}$ 2,4,6-trimethyl MA (*M. bohemicum*, *M. szulgai* and *M. intermedium*) (Torkko et al., 2001; Torkko et al., 2002; Torkko et al., 2003). The presence of
$C_{20}$ 2-methyl and $C_{21}$ 2-methyl MAs in both Censo mycobacteria and mycobacteria from the closely related *M. simiae* complex indicate that these MAs might be a common feature in the *M. simiae* complex. However, these MAs have been also found in more distantly related mycobacterial species like *M. interjectum* and *M. malmoense*, while common pathogenic mycobacteria like *M. tuberculosis* or *M. bovis* do not produce these shorter chain MAs. These pathogenic mycobacteria contain a $C_{27}$ 2,4,6-methyl MA (*M. tuberculosis*) and a $C_{26}$ 2,4-methyl (*M. bovis*) as the shortest chain MAs (Minnikin et al.,
1993a; Redman et al., 2009) which are not present in the Censo soils. Some more distantly related mycobacteria can even contain much shorter chain fatty acids like $C_{11}$ 2-methyl MA (*M. interjectum* or *M. intermedium*), $C_{15}$ 2-methyl MA (e.g. *M. kansaii* or *M. intermedium*) or $C_{16}$ 2,4-dimethyl MA (*M. gastrii and M. kansasii*) (Torkko et al., 2003), but these are not found in the Censo MA inventory.

The most unique feature that distinguishes the MAs of the Censo mycobacteria from cultivated mycobacterial
species is the occurrence of methyl groups in the middle of the fatty acid chain at positions C-12 and C-16 in $C_{22}$ 2,12-dimethyl and $C_{29}$ 2,4,6,8,16-pentamethyl MAs, respectively. To the best of our knowledge, this mid-chain methyl branching has only been reported once before, in the mycobacterial species *M. palustre,* also from the *M. simiae* complex (Torkko et al., 2002), which is closely related to the species living in the Censo soil. However, the methyl branching in *M. palustre* is at position C-9 ($C_{22}$ 2,9-dimethyl MA) (Torkko et al., 2002).

The fatty acid profile of the Censo soils shows longer chain MAs (e.g. $C_{28}$ 2,4,6,8-tetramethyl and $C_{31}$ 2,4,6,8,10-pentamethyl MAs) which are even more abundant than $C_{24}$ and $C_{26}$ long-chain n-alkyl fatty acids. This feature has not been previously reported in mycobacteria including mycobacteria from the closely-related *M. simiae* complex like *M. heidelbergense* and *M. palustre*, which synthesize much higher amounts of regular fatty acids over MAs (Torkko et al., 2002; Torkko et al., 2003). Some mycobacterial species from the *M. simiae* complex (i.e. *M. lentiflavum, M. florentinum* and
*M. genavense*) and other more distantly related mycobacteria (e.g. *M. paraffinicum* and *M. smegmatis*) (Torkko et al., 2002; Torkko et al., 2003; Fernandes and Kolattukudy, 1997; Chou et al., 1998) do not even contain MAs.



In conclusion, the MA patterns in the Censo soil mycobacteria are clearly different from those of previously cultivated mycobacterial species. This could be caused by environmental conditions near the Censo seep, which may have induced adaptions and regulation processes within the biosynthesis systems of MAs in the Censo mycobacteria or may just

be a chemotaxonomic feature. Further studies of other soils that contain mycobacteria should reveal how unique the MAs detected in the Censo soils are.

### 3.4 Role of the mycobacteria at the Censo seep

The high relative abundances of mycobacteria and MAs based on both the relative 16S rRNA gene abundance and FA

composition in the soil close to the main Censo seep (Table 1 and Fig. 2), combined with the decrease of these abundances in soils further away from the seep, hint to the potential involvement of mycobacteria in gas oxidation processes at the gas seep system. To further investigate this, the $\delta^{13}C$ values of the MAs, as well as regular fatty acids and hopanoic acids, were analyzed in the Censo seep soils (Fig. 5). The $\delta^{13}C$ values of methane (-30 to -35 ‰) and ethane (-25 ‰) in the released gases at the Censo seep have previously been reported by Grassa et al. (2004).

At the seep site, regular and unsaturated $C_{16}$, $C_{18}$, $C_{22}$ and $C_{24}$ fatty acids showed no significant depletion in their carbon isotopic composition ($\delta^{13}C$ = -25 to -30 ‰), while at 1.8 m distance these FAs feature a bit more depleted $\delta^{13}C$ values ranging from -33 to -37 ‰ (Fig. 5). As mentioned before, the $C_{16}$ and $C_{18}$ FAs could originate from Type I and Type II methanotrophs (e.g. Bowman, 2006; Dedysh et al., 2007; Bodelier et al., 2009) although larger depletion of ca. 10 to 20 ‰ relative to the methane source is generally expected for fatty acids of aerobic methanotrophs (Jahnke et al., 1999;

Blumenberg et al., 2007; Berndmeyer et al., 2013). Thus, other soil microbes using soil organic matter as a carbon source are likely to contribute to the pool of these fatty acids at Censo 0 m and 1.8 m. This agrees with the typical bulk $\delta^{13}C$ values of -25 to -30 ‰ in temperate soils (Balesdent et al., 1987; Huang et al., 1996) and the presence of saturated and unsaturated $C_{16}$ and $C_{18}$ fatty acids even further away from the seep, at 13.2 m (Fig. 2).

The $C_{32}$ 17β,21β(H)-hopanoic acid shows more depleted $\delta^{13}C$ values ranging from -42 to -48 ‰, at Censo 0 m and

1.8 m, respectively (Fig. 5), suggesting an origin from bacteria involved in the cycling of a $^{13}C$ depleted carbon source like methane at this gas seep. The $C_{32}$ hopanoic acid is a diagenetic product of bacteriohopanepolyols (Rohmer et al., 1984; Ries-Kautt and Albrecht, 1989; Farrimond et al., 2002), which could be produced by some of the aerobic methanotrophs (e.g. Methylocystaceae or Methylococcales) (Zundel and Rohmer, 1985; van Winden et al., 2012) identified in the Censo seep soils (Table 1). However, as discussed above, the mycobacteria in the soil, which are closely related to the *M. simiae*

complex (Fig. 1), might also be able to synthesize hopanoids and therefore could be contributing to the hopanoid pool.

Depleted $\delta^{13}C$ values are observed in the MAs (-38 to -48 ‰) close to the Censo seep at 0 m and 1.8 m (Fig. 5), indicating that these are likely synthesized by organisms that use a $^{13}C$-depleted carbon source rather than soil organic matter. The Censo seep releases high amounts of methane (76-86 % of total released gas) and minor amounts of higher gaseous hydrocarbons (ethane, propane etc.) as well as $CO_2$ and $N_2$ (Etiope et al., 2002; Grassa et al., 2004). Thus, it would





appear that the Censo mycobacteria are using $^{13}$C-depleted methane as their carbon source as it is the major released gas at the Censo seep. This is in agreement with the decreasing relative abundance of mycobacteria and MAs away from the main seepage according to the decrease in the released gas. Furthermore, the δ$^{13}$C values of the MAs are more negative than the δ$^{13}$C value of the released methane, as expected for methanotrophs, and dissimilar to bulk soil organic matter and the simple FAs likely derived from heterotrophic bacteria.

However, these results are not completely in agreement with previous incubation and genetic studies, which showed that mycobacteria are not able to utilize methane but rather use other gaseous hydrocarbons like ethane and propane as well as alkenes, methanol, and carbon monoxide as their carbon source (Park et al., 2003; Coleman et al., 2011; Coleman et al., 2012; Martin et al., 2014). Studies from the 1950s reported high abundances of mycobacteria in soils from areas of oil and gas production and in areas of petroliferous gas seeps, hinting to their potential involvement in gas oxidation processes

(Davis et al., 1956; Davis et al., 1959). Cultivation experiments of those soils confirmed that mycobacteria did not utilize methane but higher gaseous hydrocarbons (ethane and propane) (Davis et al., 1956; Dworkin and Foster, 1958). These results suggest that the mycobacteria in the Censo soils are perhaps not using methane, but possibly other gaseous hydrocarbons in the seep, like ethane or propane. However, it should be noted that two previous studies have described mycobacterial species *Mycobacterium flavum var. methanicum*, *Mycobacterium methanicum n. sp.* and *Mycobacterium* ID-Y

that were able to oxidize methane (Nechaeva, 1954; Reed and Dugan, 1987). Alternatively, mycobacteria at the Censo seep could act as indirect methane utilizers by using secondary products of methane oxidation performed by other methanotrophs, like methanol. Indeed, some studies have shown that cultured pathogenic mycobacteria were able to utilize methanol (Reed and Dugan, 1987; Park et al., 2003; Park et al., 2010). However, this is difficult to reconcile with their very high abundances (up to 34.1 %) compared to the low abundance of typical methanotrophs like Methylococcales or Methylocystaceae (up to

5.1 %) near the seep (Table 1).

Overall, based on the clear abundance of mycobacterial 16S rRNA sequences in the Censo seep soils, the novel $^{13}$C depleted MAs identified here may be useful biomarkers for the presence of hydrocarbon-oxidizing mycobacteria in soils. These unique MAs in combination with $^{13}$C depletion could be used to trace mycobacteria in present and past environments, specifically those influenced by hydrocarbon seepage. Longer chain fatty acids and branched fatty acids like MAs have been

shown to be more resistant than other biomolecules (e.g. short-chain fatty acids) to diagenetic changes in diverse studies of fossil forests, sediment cores from the Gulf of California, and petroleum systems (Staccioli et al., 2002; Wenger et al., 2002; Camacho-Ibar et al., 2003). Indeed, studies have indicated the presence of MAs of *M. tuberculosis* on ancient bones from a 17,000 year old bison and from a ca. 200 year old human skeletons (Redman et al., 2009; Lee et al., 2012), suggesting a high preservation potential of these lipids.

Nevertheless, future research should investigate the presence of MAs in ancient soils, soil cores, and ancient terrestrial hydrocarbon seeps. Additionally, further detailed incubation studies and genomic analysis of the Censo mycobacteria are required to elucidate the exact role of the mycobacteria in gas oxidation processes at the Censo seep.


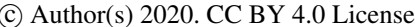

## 4. Conclusion

Soils from the Fuoco di Censo Everlasting Fire show high relative abundances (up to 34 %) of uncultivated mycobacterial 16S rRNA gene sequences. These Censo mycobacteria are phylogenetically distant from the typical pathogenic mycobacteria *Mycobacterium tuberculosis* or *M. leprae,* and more closely related to the *M. simiae* complex like *M. heidelbergense* and *M. palustre*. At the main seep, Censo soils feature a unique MA pattern especially in the longer chain MAs. The most abundant MAs were tentatively identified as 2,4,6,8-tetramethyl-tetracosanoic acid ($C_{28}$ 2,4,6,8-tetramethyl

MA) and 2,4,6,8,10-pentamethyl-hexacosanoic acid ($C_{31}$ 2,4,6,8,10-pentamethyl MA). The Censo soils also contained MAs with novel mid-chain methyl branching at positions C-12 and C-16 ($C_{22}$ 2,12-dimethyl and $C_{29}$ 2,4,6,8,16-pentamethyl MAs). The MA pattern in the Censo seep soils are clearly different from those reported for the well-studied mycobacteria like *M. tuberculosis* or *M. leprae* and from the closely related *M. simiae* complex. Only $C_{20}$ 2-methyl, $C_{21}$ 2-methyl and $C_{25}$ 2,4,6-trimethyl MAs have been found previously in other mycobacteria from the *M. simiae* complex (e.g. *M. heidelbergense*) and

three more distantly related mycobacteria (e.g. *M. interjectum*). These MAs have relatively low $\delta^{13}C$ values, suggesting that Censo mycobacteria use a carbon source depleted in $^{13}C$, such as methane, higher gaseous hydrocarbons or secondary products of gas oxidation processes, like methanol. The novel identified MAs in the Censo samples offer a new tool, besides DNA-based techniques, to investigate soils from present and past terrestrial environments for the presence of mycobacteria potentially involved in the cycling of gases.


### Data availability

Data will be made available on request to the corresponding author.

### Author contribution

NTS, DR and SS planned research. NTS, FG and CRW collected samples. MH and AJM provided the synthetic $C_{27}$
mycocerosic acid standard. NTS performed lipid analysis. LV analyzed 16S rRNA gene sequencing data. NTS, SS and LV interpreted the data. NTS wrote the paper with input from all authors.

### Competing interests

The authors declare that they have no conflict of interest.


### Acknowledgements

We thank Marianne Baas, Monique Verweij, Jort Ossebaar, Ronald van Bommel, Sanne Vreugdenhil and Maartje Brouwer for technical assistance and Marcel van der Meer for discussion about isotopic values of the fatty acids. This study received





funding through grants to LV, JSSD, and SS from the Netherlands Earth System Science Center (NESSC) and Soehngen
Institute for Anaerobic Microbiology (SIAM) through Gravitation grants (024.002.001 and 024.002.002) from the Dutch
Ministry for Education, Culture and Science.

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

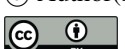



**Figure 1 Maximum likelihood (ML) phylogenetic tree of the Mycobacterial 16S rRNA gene fragments (i.e. 294 bp; in bold) generated by amplicon sequencing and representative for the two OTUs present in the soils from the Censo seep "everlasting fire". The 16S rRNA gene sequence of *Corynebacterium diphtheriae* was used as an outgroup and other Mycobacterial 16S rRNA gene sequence are plotted for reference. The ML tree is based on the General Time Reversible model with gamma distribution plus invariable sites. Mycobacterial species biosynthesizing MAs are indicated in red font, species not containing MAs are shown in blue and species for which MAs have not been analyzed are shown in black. The mycobacterial species producing MAs (in red) are labelled with their dominant MA in brackets (total carbon number, Me= methyl, x = unidentified position of methyl group).**



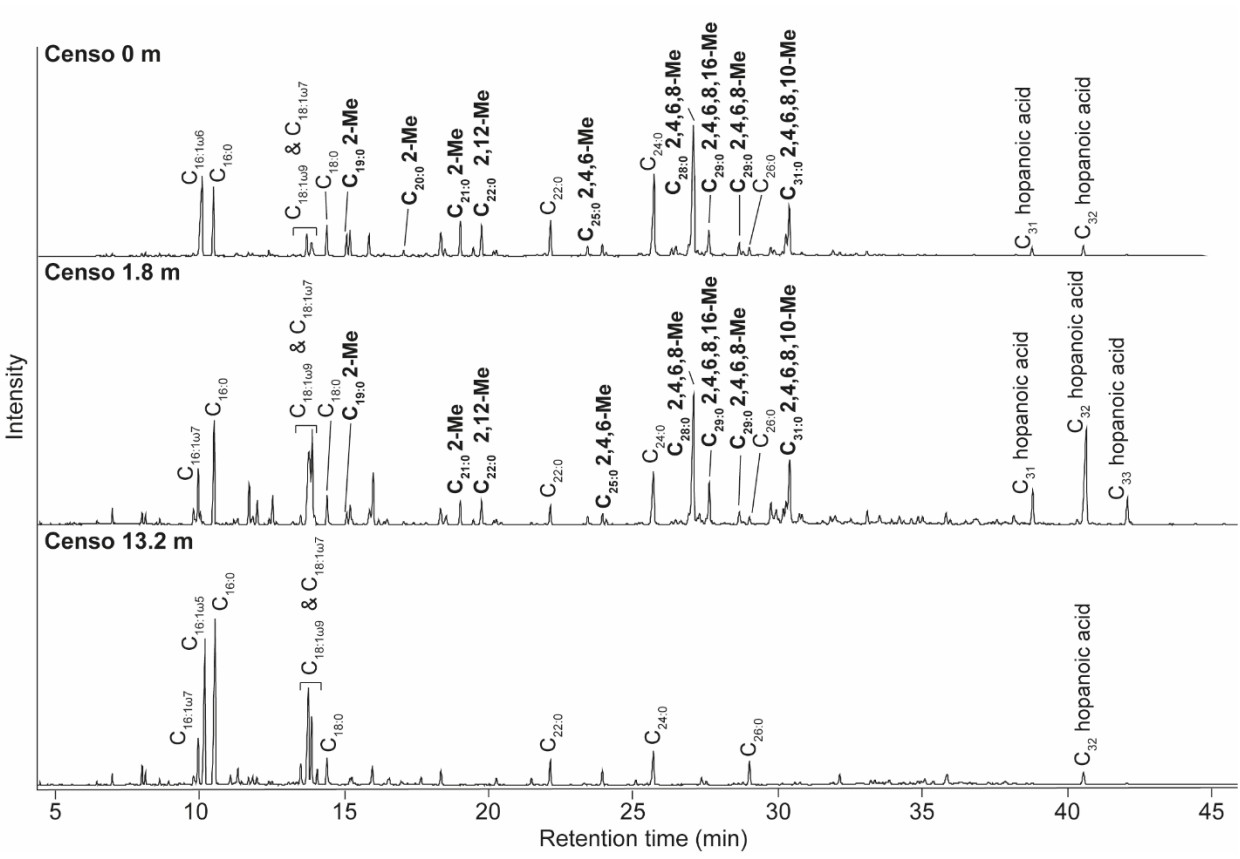

**Figure 2 Total ion chromatograms of the saponified and derivatized (BF₃) fatty acid fractions from the Censo seep soils in increasing distance from the main seepage showing the distributions of PLFAs, hopanoic acids, and MAs. The black bold annotations show the tentatively identified MAs and the carbon position of their methyl groups (Me).**





665 **Figure 3 Mass spectra of the methyl ester (left panels) and 'picolinyl'-ester (right panels) derivatized MAs of the Censo 0 m soil sample, with proposed molecular structures and fragmentation patterns. 2,12-dimethyl-eicosanoic acid (C22 2,12-Me MA) (A and B), 2,4,6,8-tetramethyl-tetracosanoic acid (C28 2,4,6,8-Me MA) (C and D), and 2,4,6,8,10-pentamethyl-hexacosanoic acid (C31 2,4,6,8,10-Me MA) (E and F). The dashed boxes show a 10 times exaggerated view into the indicated area of the mass spectrum.**





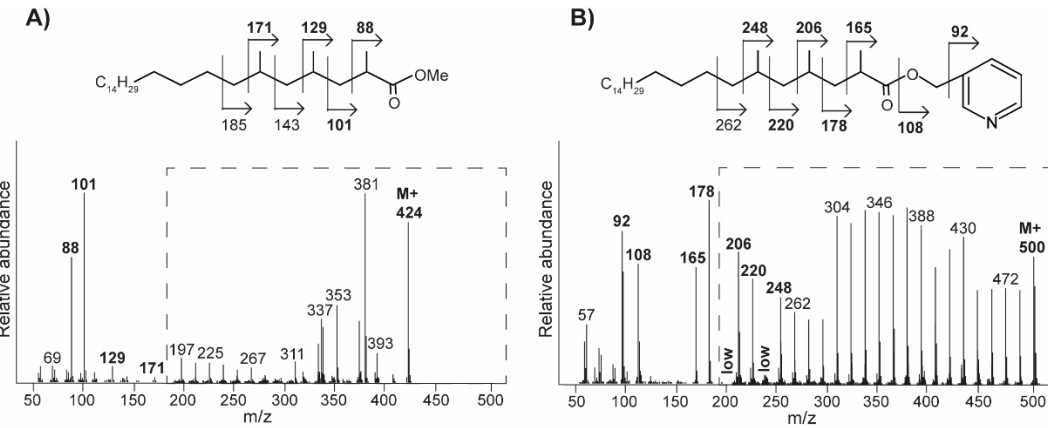

**Figure 4 Mass spectra with fragmentation and annotated molecular structures of the A) methyl ester and B) 'picolinyl' ester synthetic 2,4,6-trimethyl-tetracosanoic acid ($C_{27}$ 2,4,6-Me MA standard). The dashed boxes show a 10 times exaggerated view into the indicated area of the mass spectrum.**

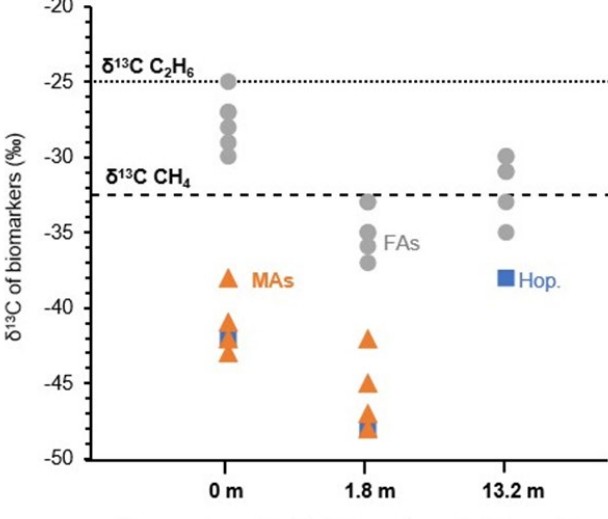

**Figure 5 The stable carbon isotopic composition ($\delta^{13}$C) of biomarkers in the Censo soils at increasing distance from the main gas seepage. Biomarkers shown are fatty acids (FAs = grey circle), mycocerosic acids (MAs = orange triangle), and the $C_{32}$ hopanoic acid (Hop. = blue square). Data points represent the mean average of two analysis. The $\delta^{13}$C values of the released methane (CH$_4$ ~ -32.5 ‰) and ethane (C$_2$H$_6$ = -25 ‰) are indicated by dashed lines in the plot.**



**Table 1 Distribution of the main microbial groups (relative abundance % of assigned reads) based on 16S rRNA gene amplicon sequencing at three distances from the main gas seep in the Censo soils.**

685

|  | 0 m | 1.8 m | 13.2 m |
|---|---|---|---|
| Archaea; Euryarchaeota | 53.5 | 0.0 | 0.0 |
| Archaea; Thaumarchaeota | 0.0 | 0.0 | 2.5 |
| Bacteria; Acidobacteria | 0.2 | 3.0 | 10.5 |
| Bacteria; Actinobacteria | 36.8 | 23.6 | 46.0 |
| Acidimicrobiales, other | 0.0 | 0.3 | 2.3 |
| **Corynebacteriales, Mycobacteriaceae, Mycobacterium** | **34.1** | **8.5** | **0.7** |
| Frankiales, Geodermatophilaceae, Geodermatophilus | 0.0 | 5.1 | 0.1 |
| Micrococcales, Microbacteriaceae, Humibacter | 0.1 | 3.0 | 0.0 |
| Micromonosporales, Micromonosporaceae, Micromonospora | 1.0 | 0.0 | 0.1 |
| Pseudonocardiales, Pseudonocardiaceae, Pseudonocardia | 0.0 | 0.0 | 1.6 |
| Rubrobacterales, Rubrobacteriaceae, Rubrobacter | 0.0 | 0.0 | 6.2 |
| Gaiellales, Gaiellaceae, Gaiella | 0.0 | 0.0 | 5.1 |
| Solirubrobacterales, 288-2, uncultured bacterium | 0.0 | 0.0 | 2.2 |
| Solirubrobacterales, Elev-16S-1332, uncultured bacterium | 0.0 | 0.6 | 4.3 |
| Solirubrobacterales, Solirubrobacteraceae, Solirubrobacter | 0.0 | 0.0 | 1.8 |
| others | 1.6 | 6.0 | 21.5 |
| Bacteria; Armatimonadetes | 0.0 | 0.0 | 0.2 |
| Bacteria; Bacteroidetes | 0.3 | 1.2 | 2.1 |
| Bacteria; Chloroflexi | 0.2 | 30.5 | 10.5 |
| Anaerolineales, Anaerolineae, Anaerolineales | 0.0 | 7.9 | 0.2 |
| Ktedonobacteria | 0.1 | 17.8 | 0.1 |
| Bacteria; Firmicutes | 5.2 | 15.7 | 0.6 |
| Bacteria; Gemmatimonadetes | 0.0 | 0.0 | 3.1 |
| Bacteria; Latescibacteria | 0.0 | 0.0 | 0.2 |
| Bacteria; Nitrospirae | 0.0 | 0.0 | 0.2 |
| Bacteria; Planctomycetes | 0.2 | 1.5 | 3.7 |
| Bacteria; Proteobacteria | 2.9 | 21.1 | 17.2 |
| Alphaproteobacteria, Rhizobiales, Methylobacteriaceae | 0.0 | 0.3 | 0.8 |
| Alphaproteobacteria, Rhizobiales, Methylocystaceae | 0.0 | 4.2 | 0.0 |



| | | | |
|---|---|---|---|
| Gammaproteobacteria, Methylococcales | 0.2 | 5.1 | 0.2 |
| Bacteria; Saccharibacteria | 0.1 | 2.1 | 0.2 |
| Bacteria; Tectomicrobia | 0.0 | 0.0 | 0.7 |
| Bacteria; Verrucomicrobia | 0.3 | 0.3 | 1.7 |
| Verrucomicrobia Incertae Sedis, Candidatus Methylacidiphilum | 0.2 | 0.0 | 0.0 |
| **Number of reads assigned** | **140,206** | **63,916** | **259,714** |

**Table 2 Chemical variability and occurrence of MAs in the Censo seep soils and in the most relevant mycobacterial species. The underlined names of the mycobacterial species indicate the major MA configuration in the mycobacterial species. The x in the position of methylations in the *n*-alkyl chain features an unidentified position of the methyl group. MAs indicated in bold typeface are MAs identified in the Censo seep soils**

| Chemical structure of MAs | | Occurrence |
|---|---|---|
| **Length of *n*-alkyl chain** | **Position of methyl group(s)** | |
| $C_{16}$ | 2,4 | *M. marinum* [1] |
| **$C_{18}$** | **2** | **Censo** |
| $C_{19}$ | 2 | **Censo,** *M. palustre* [2] |
| **$C_{20}$** | **2** | **Censo,** *M. bohemicum, M. heidelbergense, M. malmoense, M. interjectum, M. palustre* [1,2,3,4,5] |
| $C_{20}$ | 2,4 | *M. asiaticum, M. szulgai, M. intermedium, M. heidelbergense, M. malmoense* [1,2,4,5] |
| $C_{20}$ | 2,9 | *M. palustre* [2] |
| **$C_{20}$** | **2,12** | **Censo** |
| $C_{20}$ | 2,4,6,x | *M. botniense* [4] |
| **$C_{22}$** | **2,4,6** | **Censo,** *M. bohemicum, M. szulgai, M. intermedium* [1,2,4] |
| $C_{22}$ | 2,4,6,x,x | *M. botniense* [4] |
| $C_{24}$ | 2,4 | *M. bovis* [6,7] |
| $C_{24}$ | 2,4,6 | *M. tuberculosis, M. bovis, M. kansasii, M. marinum, M. ulcerans, M. bohemicum, M. heidelbergense, M. malmoense, M. interjectum* [2,3,4,5,6,7,8] |





| $C_{24}$ | **2,4,6,8** | **_Censo_** |
|---|---|---|
| $C_{26}$ | 2,4,6 | _M. tuberculosis, M. leprae_, M. bovis, _M. kansasii, M. marinum, M. ulcerans_ [6,7] |
| $C_{24}$ | **2,4,6,8,16** | **Censo** |
| $C_{25}$ | **2,4,6,8** | **Censo** |
| $C_{26}$ | 2,4,6,8 | _M. tuberculosis, M._ leprae, _M. bovis, M. kansasii, M. marinum, M. ulcerans_ [6,7,8,9,10] |
| $C_{26}$ | **2,4,6,8,10** | **Censo** |
| $C_{28}$ | 2,4,6,8 | _M. tuberculosis, M. leprae, M. bovis, M. kansasii, M. marinum_ [6,7,8,11] |
| $C_{28}$ | 2,4,6,8,10 | _M. tuberculosis, M. leprae, M. kansasii_ [6,7] |
| $C_{30}$ | 2,4,6,8 | _M. leprae_ [6] |

[1]Chou et al. (1996); [2]Torkko et al. (2002), [3]Torkko et al. (2001), [4]Torkko et al. (2003), [5]Valero-Guillén et al. (1988), [6]Minnikin et al. (1993a), [7]Minnikin et al. (1985), [8]Daffé and Laneelle (1988), [9]Donoghue et al. (2017), [10]Redman et al. (2009), [11]Minnikin et al. (2002).