# Peer review of "Novel hydrocarbon-utilizing soil mycobacteria synthesize unique mycocerosic acids at a Sicilian everlasting fire"

_Biogeosciences, 2020_

## Referee Comment (RC1) · Rienk Smittenberg (Referee) · 24 Nov 2020

The paper by Smit et al is a fine example of the ongoing quest to identify new lipid biomarkers from the environment, and linking these with their parent organisms. In this case, the authors specifically went to sample soils near a hydrothermal vent where they identified raised levels of Mycobacteria, a group most known for their pathogeny like Tuberculosis, but also for their capacity to grow on gaseaous hydrocarbons. They used elegant manners to identify (tentatively until possibly some future NMR work) a range of unusual 'Mycoseroric' fatty acids that appear to be specific for these environments, on top they also investigated the 13C content of these lipids further substantiate whether

methane and/or ethane acts as the carbon source. With this paper the field of organic geochemistry is again a little richer in the amount of diagnostic environmental lipids. The paper is well written and I could not detect any errors.

---

## Referee Comment (RC2) · Gordon Inglis (Referee) · 3 Dec 2020

In this paper, Nadine Smit and co-authors investigate the lipid biomarker inventory in soils near a gas seep. They find a very high abundance of mycobacteria and 13C-depleted mycocerosic acids (MAs) near to the seep. The abundance of mycobacteria decreases away from the seep. This is accompanied by a corresponding increase in the carbon isotope composition of MA's. This implies that mycobacteria are utilising 13C-depleted substrates and demonstrates that MA's have the potential to provide new insights into carbon cycling within gas-rich environments.

This is a really lovely study with very few faults. The combined 16S rRNA/biomarker

approach is novel and the authors did an impressive job with the structural identification. I was also glad to see mass spectra in the figures – this will be very useful for other researchers! Overall, a really nice, well-written paper which should be published in Biogeosciences.

I do have a couple of suggestions:

1) You appear to have a unique, source-specific biomarker which can be used to study gas oxidation processes. Very cool! This is a useful addition to our "biomarker toolkit" and complements other "gas oxidation" proxies (e.g. BHPs or fatty acids which are specific to methanotrophs, or hopanoid carbon isotopes values which reflect the balance between heterotrophy and methanotrophy). However, you only identified MA's very close to the gas seep and they were absent in the 13m soil. Therefore, it is plausible that we might never find these lipids in paleo-record (at least using conventional GC-MS). Or am I wrong? Are there other sites/settings where these might be likely to occur? Perhaps in the marine realm? e.g. hydrothermal vent systems, mud volcanoes, cold seeps etc.

2) You mentioned that MA's have been found in ancient bones (ca, 20,000 years ago) but is there the potential for these lipids to be preserved further back in geological time? i.e. millions of year? And what putative degradation products might we expect to find in the geological record?

3) Did you look at the non-fatty acid fractions? Could there be any other diagnostic lipids hiding in the aliphatic fraction (for example)?

Minor comments:

Do we know why mycobacteria synthesise MA's with this weird branching pattern?

Have MA's been identified in any other bacterial strains? Or have people never looked?

Figure 3: although the abundance of mycobacteria are highest near the seep, the lowest del13C values actually occur 1.8m from the seep. One might (incorrectly) assume

that the lowest values would occur directly at the seep. Is there a suitable explanation for this?

Figure 2: I am not surprised that the C32 hopanoic acid dominates the non-seep samples – it appears to be the dominant hopanoic acid elsewhere (see Inglis 2018; https://www.sciencedirect.com/science/article/pii/S0016703718300036). However, I was intrigued by the C31 and C33 hopanoic acids. There doesn't appear to be any mention of these in the text – I wondered if: 1) you obtained del13C values from these lipids, and 2) what the putative source of these lipids are?

L39: I would also mention here (and elsewhere) that you can use: 1) specific bacteriohopanepolyols for methane oxidation (e.g. Talbot et al., 2016 OG; van Winden et al. 2012 GCA) and 2) hopanoid carbon isotopes (Inglis et al 2019 GCA, van Winden 2020 Geobiology) to probe methanotrophy in terrestrial environments.

L95: Sample collection = why 0.8 and 13.2m? Any methodological reasoning?

L339: alternatively, the -33 to -37 per mil values could reflect a mixed bacterial community (e.g. heterotrophs + methanotrophs; e.g. Inglis et al., 2019 GCA).

L350: apart from having Shc gene, any other evidence the mycobacteria synthesise hopanoids? Any existing cultures?

---

## Editor Comment (EC1) · Sebastian Naeher (Editor) · 6 Dec 2020

This is a very interesting study and well written manuscript. In addition to the comments raised by the reviewers, I would also like to add a few more, very minor comments for you to consider:

Line 14: Could you please add more examples here instead of stating "various habitats" of mycobacteria. I see that this information is provided in the main text of the manuscript, but thought it would be good to add some of this information here to show that these organisms have a high diversity.

Line 25 and elsewhere: Better add full name (Fuoco di Censo instead of Censo)?

Line 99-109: Could you please note why you use BDE for extraction and saponification instead of other methods? Using BDE is a great method to prevent decomposition of some lipids, but then you undertake base hydrolysis? Did you also make a comparison of your results by looking at intact polar lipids/PLFAs as well? For the saponification step, is there a reason why you adjusted the pH to 5 and not lower than that?

Line 118: polar compounds

Lines 197-198: Are you able to you add an approximate number to show how much/percentage the uncultured mycobacteria represent?

Line 211: However instead of although?

Lines 217-219: Could you please add more details about these "explorative searches"? Or even a reference to provide more information about the synthesis of biohopanoids in mycobacteria?

Line 244: consider changing to methyl group (twice in this line)

Lines 250-251: M+ with superscript "+" and remove "of m/z", also add respectively at end of sentence.

Line 274: Biomarkers

Line 333: Could you note briefly why the d13C values of methane and ethane are quite enriched in these seeps? Even if the background information is surely provided in the cited study and known to many readers of your manuscript, I think it could be useful for the reader that is not very familiar with this, because methane is commonly highly depleted in other settings.

Line 387: Not necessarily limited to this seep/site. Investigating various other, similar settings would also provide more insight into the abundance and community composition of mycobacteria and their lipid composition (relative amounts and stable isotope

composition), as well as to better understand how these lipids help mycobacteria to adapt to these special environments.

---

## Author Comment (AC1) · 9 Dec 2020

We thank the referee, Dr. Smittenberg, for taking the time to consider our manuscript and for his very positive assessment of the manuscript.
* * *

---

## Author Comment (AC2) · 9 Dec 2020

We thank the referee, Dr. Inglis, for taking the time to consider our manuscript, his positive assessment and the helpful comments. Below, we respond to the detailed comments.

1) You appear to have a unique, source-specific biomarker which can be used to study gas oxidation processes. Very cool! This is a useful addition to our "biomarker toolkit" and complements other "gas oxidation" proxies (e.g. BHPs or fatty acids which are specific to methanotrophs, or hopanoid carbon isotopes values which reflect the balance between heterotrophy and methanotrophy). However, you only identified MA's

very close to the gas seep and they were absent in the 13m soil. Therefore, it is plausible that we might never find these lipids in paleo-record (at least using conventional GC-MS). Or am I wrong? Are there other sites/settings where these might be likely to occur? Perhaps in the marine realm? e.g. hydrothermal vent systems, mud volcanoes, cold seeps etc.

Thanks for the question. Indeed, the mycocerosic acids are lacking at 13 m, a strong indication that they are associated with gas oxidation. The control site is 13 m away and downhill from the seepage area, so it was not surprising to find no mycocerosic acids as there is no significant gas or oil emission and very low abundance of sequences of mycobacteria. Thus, these biomarkers seem really prevalent near seep sites, similar to biomarker lipids of aerobic and anaerobic methane oxidizers which are also abundant in the geological record. We therefore expect them to be present in other environmental settings where a lot of gas or oil is available, e.g. mud volcanoes, cold or petroleum seeps, or thawing permafrost. Furthermore, sequences of mycobacteria have been found to be abundant in a wide variety of different (including marine) environments, but these studies never examined, to the best of our knowledge, for the presence of mycocerosic acids. The report of these compounds in this manuscript might therefore hopefully lead to recognition of these unique lipids in other environmental settings.

2) You mentioned that MA's have been found in ancient bones (ca, 20,000 years ago) but is there the potential for these lipids to be preserved further back in geological time? i.e. millions of year? And what putative degradation products might we expect to find in the geological record?

As mentioned in the last section of the manuscript (i.e. 3.4), we expect that the preservation potential of MAs is similar as general fatty acids (C16 and C18 FAs). Under the right conditions fatty acids may be preserved in ancient sediments up to the Miocene (e.g. Ahmed, M., Schouten, S., Baas, M., De Leeuw, J., 2001. Bound lipids in kerogens from the Monterey Formation, Naples Beach, California. The Monterey Formation: From Rock to Molecules. Columbia University Press, New York, 189-205.). Furthermore, there may be diagenetic products formed from mycocerosic acids, such as shorter chain FAs or hydrocarbons. This would be very interesting to study in more detail in future projects.

3) Did you look at the non-fatty acid fractions? Could there be any other diagnostic lipids hiding in the aliphatic fraction (for example)?

Yes, we did scan other fractions and the total lipid extract for IPLs for the Censo samples. We did find some unknown compounds which potentially could be connected to the mycocerosic acids but at this point the results are still inconclusive and not suitable for publication yet.

- Do we know why mycobacteria synthesise MA's with this weird branching pattern? - Have MA's been identified in any other bacterial strains? Or have people never looked?

The current state of knowledge is that mycocerosic acids are unique to mycobacteria to synthesize these multi-methyl branched fatty acids. Mycobacteria possess two FA biosynthesis systems type I (eukaryotic type) and type II (prokaryotic type) to produce their lipid inventory which is in a double cell membrane containing remarkably long FAs with up to 90 carbon atoms. The MAs are synthesized by FAS type I with a methyl malonyl CoA instead of the malonyl CoA found in usual FAs (e.g. 4 rounds of C18 extension); this gene is known as the mas gene (mycocerosic acid synthase). The MA lipids make the mycobacterial cell membrane extremely hydrophobic and impermeable which allows them to be resistant against toxins etc. and colonize at a lot of nutrient rich interfaces. We will address this in somewhat more detail in the revised manuscript. Please see for further info for example:

Brennan, P.J., 2003. Structure, function, and biogenesis of the cell wall of Mycobacterium tuberculosis. Tuberculosis 83, 91-97.

Gago, G., Diacovich, L., Arabolaza, A., Tsai, S.-C., Gramajo, H., 2011. Fatty acid biosynthesis in actinomycetes. FEMS Microbiology Reviews 35, 475-497.

- Figure 3: although the abundance of mycobacteria are highest near the seep, the lowest del13C values actually occur 1.8m from the seep. One might (incorrectly) assume that the lowest values would occur directly at the seep. Is there a suitable explanation for this?

Unfortunately, it is not completely clear why we see these results. One explanation could be that it depends on the gas flux and how they adapt to this. It might be that mycobacteria can better utilize lower gas concentrations. We know about high and low affinity methanotrophs, and possibly these mycobacteria behave similar and are acting as high affinity methanotrophs.

- Figure 2: I am not surprised that the C32 hopanoic acid dominates the non-seep samples – it appears to be the dominant hopanoic acid elsewhere (see Inglis 2018; https://www.sciencedirect.com/science/article/pii/S0016703718300036). However, I was intrigued by the C31 and C33 hopanoic acids. There doesn't appear to be any mention of these in the text – I wondered if: 1) you obtained del13C values from these lipids, and 2) what the putative source of these lipids are?

We assume that the C31 and C33 hopanoic acids might have similar sources as the C32 hopanoic acids, as we shortly mentioned in section 3.2. Potential sources could be a range of bacteria like Alpha- and Gammaproteobacteria, Planctomycetes and Acidobacteria. We did obtain del13C values for them; they showed the same depletion as the C32 hopanoic acid with del13C values ranging from -40 to -50 ‰.The del13C values of these two hopanoic acids were not added in the manuscript since we felt that it did not add much value to the main story of the mycocerosic acids.

We will address all issues raised, where relevant, in a revised manuscript.

---

## Author Comment (AC3) · 9 Dec 2020

We thank the associate editor, Dr. Naeher, for taking the time to consider our manuscript and provide helpful questions and suggestions for improvement. Below, we respond to the detailed comments.

- Line 99-109: Could you please note why you use BDE for extraction and saponification instead of other methods? Using BDE is a great method to prevent decomposition of some lipids, but then you undertake base hydrolysis? Did you also make a comparison of your results by looking at intact polar lipids/PLFAs as well? For the saponification step, is there a reason why you adjusted the pH to 5 and not lower than that?

We used the BDE extraction method to extract as many compound classes (including intact polar lipids) as possible out of the Censo soils. Aliquots of our lipid extracts were used for different types of analysis, such as analysis of bacteriohopanepolyols and intact polar lipids which require BDE extraction. The results of these ongoing analyses will be presented in follow-up research papers. The focus of this study is the identification of the unique mycocerosic acids and base hydrolysis (applied to release fatty acids from much more structurally complex intact polar lipids) was the method of choice for this work. We adjusted the pH to 5 since in our experience we get the best yields for PLFA turnover using this pH value in our lab. Please see also previous studies from our lab for example:

Heinzelmann, S.M., Bale, N.J., Villanueva, L., Sinke-Schoen, D., Philippart, C.J.M., Sinninghe Damsté, J.S., Schouten, S., van der Meer, M.T.J., 2016. Seasonal changes in the D/H ratio of fatty acids of pelagic microorganisms in the coastal North Sea. Biogeosciences 13, 5527-5539.

- Lines 197-198: Are you able to you add an approximate number to show how much/ percentage the uncultured mycobacteria represent?

The relative abundances of assigned reads in the 16S rRNA gene amplicon sequencing is 34.1 % (Censo 0m) and 8.5 % (Censo 1.8m). This is listed already in Table 1, but we will include these numbers in the text in the revised manuscript.

- Lines 217-219: Could you please add more details about these "explorative searches"? Or even a reference to provide more information about the synthesis of biohopanoids in mycobacteria? In combination with comment reviewer (G. Inglis): L350: apart from having Shc gene, any other evidence the mycobacteria synthesise hopanoids? Any existing cultures?

Unfortunately, we cannot provide more detailed information as the analysis of the Censo mycobacteria metagenome was not part of this study. However, preliminary BLAST search for the shc gene in M. simiae mycobacteria showed the potential for

hopanoid production, while other more explored mycobacteria like M. tuberculosis do not possess these genes. Moreover, pathogenic mycobacteria are able to synthesize sterols instead of hopanoids as described in these papers:

Lamb, D.C., Kelly, D.E., Manning, N.J., Kelly, S.L., 1998. A sterol biosynthetic pathway in Mycobacterium. FEBS letters 437, 142-144.

Podust, L.M., Poulos, T.L., Waterman, M.R., 2001. Crystal structure of cytochrome P450 14$\alpha$-sterol demethylase (CYP51) from Mycobacterium tuberculosis in complex with azole inhibitors. Proceedings of the National Academy of Sciences 98, 3068-3073.

- Line 333: Could you note briefly why the d13C values of methane and ethane are quite enriched in these seeps? Even if the background information is surely provided in the cited study and known to many readers of your manuscript, I think it could be useful for the reader that is not very familiar with this, because methane is commonly highly depleted in other settings.

The information is provided in more detail in section 2.2 on the study area. Methane and ethane derive from overmature marine source rocks and are of thermogenic origin resulting in a very dry gas with C1/(C2+C3) ratios >100 (Grassa et al., 2004). The thermogenic origin of these gases is responsible for the enriched $\delta$13C values at the Censo seep. Hence, in contrast to other seeps, microbial sources for methane and ethane are not important. We will add this info in the revised manuscript.

We will address the reviewers and editor comments in detail in a revised manuscript.

---

## Author Response (AR1)

Title: Novel hydrocarbon-utilizing soil mycobacteria synthesize unique mycocerosic acids at a Sicilian everlasting fire
Author(s): Nadine T. Smit et al.
MS No.: bg-2020-349
MS type: Research article
Biogeosciences
Rebuttal

Dear Dr. Sebastian Naeher,

We would like to thank you, and the reviewers Dr. Rienk Smittenberg and Dr. Gordon Inglis, for taking the time to consider our manuscript and provide helpful comments and suggestions for its improvement. We are pleased that you and reviewers agreed on the appeal of our work to the readers of Biogeosciences.

We have revised the manuscript according to your comments and those of the reviewers (repeated in italic font) of which you will find a point-by-point response below.

We look forward to hear from you regarding our resubmission.

On behalf of all authors,

Nadine Smit

Associate Editor's comments (Dr. Sebastian Naeher):

*This is a very interesting study and well written manuscript. In addition to the comments raised by the reviewers, I would also like to add a few more, very minor comments for you to consider:*

Thank you for your comments and suggestions as well as your positive assessment of the manuscript. We considered them carefully and revised the manuscript accordingly.

*Line 14: Could you please add more examples here instead of stating "various habitats" of mycobacteria. I see that this information is provided in the main text of the manuscript but thought it would be good to add some of this information here to show that these organisms have a high diversity.*
We added this info to the text.

*Line 25 and elsewhere: Better add full name (Fuoco di Censo instead of Censo)?*
We changed Censo to Fuoco di Censo in the abstract. Throughout the manuscript, we prefer to use only Censo as an abbreviation for a more concise language. We introduce this abbreviation in section 2.1 study area (please see lines 84ff).

*Line 99-109: Could you please note why you use BDE for extraction and saponification instead of other methods? Using BDE is a great method to prevent decomposition of some lipids, but then you undertake base hydrolysis? Did you also make a comparison of your results by looking at intact polar lipids/PLFAs as well? For the saponification step, is there a reason why you adjusted the pH to 5 and not lower than that?*

We used the BDE extraction method to extract as many compound classes (including intact polar lipids) as possible out of the Censo soils. Aliquots of our lipid extracts were used for different types of analysis, such as analysis of bacteriohopanepolyols and intact polar lipids which require BDE extraction. The results of these ongoing analyses will be presented in follow-up research papers. The focus of this study is the identification of the unique mycocerosic acids and base hydrolysis (applied to release fatty acids from much more structurally complex intact polar lipids) was the method of choice for this work.
We adjusted the pH to 5 since in our experience we get the best yields for PLFA recovery using this pH value. Please see also previous studies from our group for example:

Heinzelmann, S.M., Bale, N.J., Villanueva, L., Sinke-Schoen, D., Philippart, C.J.M., Sinninghe Damsté, J.S., Schouten, S., van der Meer, M.T.J., 2016. Seasonal changes in the D/H ratio of fatty acids of pelagic microorganisms in the coastal North Sea. Biogeosciences 13, 5527-5539.

We added some additional info regarding these comments to the manuscript. Please see lines 107 and 114.

*Line 118: polar compounds*
This has been changed as requested.

*Lines 197-198: Are you able to you add an approximate number to show how much/ percentage the uncultured mycobacteria represent?*

The relative abundances of assigned reads in the 16S rRNA gene amplicon sequencing of uncultured mycobacteria is up to 34.1 % (Censo 0 m) in the Censo soils. This info is given in Table 1 as well as in the text in lines 192ff. We added this info also to lines 207ff as requested.

*Line 211: However instead of although?*
This has been changed as requested.

*Lines 217-219: Could you please add more details about these "explorative searches"? Or even a reference to provide more information about the synthesis of biohopanoids in mycobacteria?*
*In combination with comment reviewer (G. Inglis):* *L350: apart from having Shc gene, any other evidence the mycobacteria synthesise hopanoids? Any existing cultures?*

Unfortunately, we cannot provide more detailed information as the analysis of the Censo mycobacteria metagenome was not part of this study. However, preliminary BLAST search for the shc gene in the *Mycobacterium simiae* genome showed the potential for hopanoid production, while other more explored mycobacteria like *M. tuberculosis* do not possess these genes. Instead, pathogenic mycobacteria are able to synthesize sterols instead of hopanoids as described in these papers:

Lamb, D.C., Kelly, D.E., Manning, N.J., Kelly, S.L., 1998. A sterol biosynthetic pathway in Mycobacterium. FEBS letters 437, 142-144.

Podust, L.M., Poulos, T.L., Waterman, M.R., 2001. Crystal structure of cytochrome P450 14α-sterol demethylase (CYP51) from Mycobacterium tuberculosis in complex with azole inhibitors. Proceedings of the National Academy of Sciences 98, 3068-3073.

We added some more info and references regarding these comments to the manuscript. Please see lines 230-234.

*Line 244: consider changing to methyl group (twice in this line)*
This has been changed as requested.

*Lines 250-251: M+ with superscript "+" and remove "of m/z", also add respectively at end of sentence.*
This has been changed as requested.

*Line 274: Biomarkers*
This has been changed as requested.

*Line 333: Could you note briefly why the d13C values of methane and ethane are quite enriched in these seeps? Even if the background information is surely provided in the cited study and known to many readers of your manuscript, I think it could be useful for the reader that is not very familiar with this, because methane is commonly highly depleted in other settings.*
The information is provided in more detail in section 2.2 on the study area. Methane and ethane derive from overmature marine source rocks and are of thermogenic origin resulting in a very dry gas with C1/(C2+C3) ratios >100 (Grassa et al., 2004). The thermogenic origin of these gases is responsible for the enriched δ13C values at the Censo seep. Hence, in contrast to other seeps, microbial sources for methane and ethane are not important. We added this info here to clarify the origin of the gases (please see lines 348ff)

*Line 387: Not necessarily limited to this seep/site. Investigating various other, similar settings would also provide more insight into the abundance and community composition of mycobacteria and their lipid composition (relative amounts and stable isotope composition), as well as to better understand how these lipids help mycobacteria to adapt to these special environments.*
We added some info and changed the text regarding your suggestions (please see lines 403-408).

Reviewer 1 comments (Dr. Rienk Smittenberg):

*The paper by Smit et al is a fine example of the ongoing quest to identify new lipid biomarkers from the environment and linking these with their parent organisms. In this case, the authors specifically went to sample soils near a hydrothermal vent where they identified raised levels of Mycobacteria, a group most known for their pathogeny like Tuberculosis, but also for their capacity to grow on gaseous hydrocarbons. They used elegant manners to identify (tentatively until possibly some future NMR work) a range of unusual 'Mycoseroric' fatty acids that appear to be specific for these environments, on top they also investigated the 13C content of these lipids further substantiate whether methane and/or ethane acts as the carbon source. With this paper the field of organic geochemistry is again a little richer in the amount of diagnostic environmental lipids. The paper is well written and I could not detect any errors.*

We thank Dr. Smittenberg, for taking the time to consider our manuscript and for his very positive assessment of the manuscript.

Reviewer 2 comments (Dr. Gordon Inglis):

*In this paper, Nadine Smit and co-authors investigate the lipid biomarker inventory in soils near a gas seep. They find a very high abundance of mycobacteria and 13Cdepleted mycocerosic acids (MAs) near to the seep. The abundance of mycobacteria decreases away from the seep. This is accompanied by a corresponding increase in the carbon isotope composition of MA's. This implies that mycobacteria are utilising 13C-depleted substrates and demonstrates that MA's have the potential to provide new insights into carbon cycling within gas-rich environments. This is a really lovely study with very few faults. The combined 16S rRNA/biomarker approach is novel and the authors did an impressive job with the structural identification. I was also glad to see mass spectra in the figures – this will be very useful for other researchers! Overall, a really nice, well-written paper which should be published in Biogeosciences.*

We thank the referee for his comments and suggestions as well as his positive assessment of our manuscript. We considered them carefully and revised the manuscript accordingly.

*1) You appear to have a unique, source-specific biomarker which can be used to study gas oxidation processes. Very cool! This is a useful addition to our "biomarker toolkit" and complements other "gas oxidation" proxies (e.g. BHPs or fatty acids which are specific to methanotrophs, or hopanoid carbon isotopes values which reflect the balance between heterotrophy and methanotrophy). However, you only identified MA's very close to the gas seep and they were absent in the 13m soil. Therefore, it is plausible that we might never find these lipids in paleo-record (at least using conventional GC-MS). Or am I wrong? Are there other sites/settings where these might be likely to occur? Perhaps in the marine realm? e.g. hydrothermal vent systems, mud volcanoes, cold seeps etc.*

Thanks for the question. Indeed, the mycocerosic acids are lacking at 13 m, a strong indication that they are associated with gas oxidation as indicated in our manuscript. The control site is 13 m away and downhill from the seepage area, so it was not surprising to find no mycocerosic acids as there is no significant gas or oil emission and very low abundance of sequences of mycobacteria (please see Fig. 2 and 5 or Table 1). Thus, these biomarkers seem really prevalent near seep sites, similar to biomarker lipids of aerobic and anaerobic methane oxidizers which are also abundant in the geological record. We therefore expect them to be present in other environmental settings where a lot of gas or oil is available, e.g. mud volcanoes, cold or petroleum seeps, or thawing permafrost. Furthermore, sequences of mycobacteria have been found to be abundant in a wide variety of different (including marine) environments, but these studies never examined, to the best of our knowledge, for the presence of mycocerosic acids. The report of these compounds in this manuscript might, therefore, hopefully lead to recognition of these unique lipids in other environmental settings. We changed the last paragraph of section 3.4 according to these comments in combination with the comments from the Associate Editor. Please see lines 403 to 408.

*2) You mentioned that MA's have been found in ancient bones (ca, 20,000 years ago) but is there the potential for these lipids to be preserved further back in geological time? i.e. millions of year? And what putative degradation products might we expect to find in the geological record?*

As mentioned in the last section of the manuscript (i.e. 3.4), we expect that the preservation potential of MAs is similar as general fatty acids ($C_{16}$ and $C_{18}$ FAs). Under the right conditions fatty acids may be preserved in ancient sediments up to the Miocene (e.g. Ahmed, M.,

Schouten, S., Baas, M., De Leeuw, J., 2001. Bound lipids in kerogens from the Monterey Formation, Naples Beach, California. The Monterey Formation: From Rock to Molecules. Columbia University Press, New York, 189-205.). Furthermore, there may be diagenetic products formed from mycocerosic acids, such as shorter chain branched FAs or branched hydrocarbons. This would be very interesting to study in more detail in future projects. We added a sentence and the reference to clarify these questions (please see lines 399ff).

*3) Did you look at the non-fatty acid fractions? Could there be any other diagnostic lipids hiding in the aliphatic fraction (for example)?*

Yes, we did scan other fractions and the total lipid extract for IPLs for the Censo samples. We did find some unknown compounds which potentially could be connected to the mycocerosic acids but at this point the results are still inconclusive and not suitable for publication yet. We describe the known structures of mycobacterial lipids in the introduction (please see lines 64-75).

*Minor comments:*

*Do we know why mycobacteria synthesise MA's with this weird branching pattern?*
*Have MA's been identified in any other bacterial strains? Or have people never looked?*

The current state of knowledge is that mycocerosic acids are unique to mycobacteria to synthesize these multi-methyl branched fatty acids. Mycobacteria possess two FA biosynthesis systems type I (eukaryotic type) and type II (prokaryotic type) to produce their lipid inventory which is in a double cell membrane containing remarkably long FAs with up to 90 carbon atoms. The MAs are synthesized by FAS type I with a methyl malonyl CoA instead of the malonyl CoA found in usual FAs (e.g. 4 rounds of $C_{18}$ extension); this gene is known as the *mas* gene (mycocerosic acid synthase). The MA lipids make the mycobacterial cell membrane extremely hydrophobic and impermeable which allows them to be resistant against toxins etc. and colonize at a lot of nutrient rich interfaces.
We added more info and references on this to the manuscript in the introduction (please see lines 65-72) as well as to section 3.3 (please see line 288).

Please see for further info for example:

Brennan, P.J., 2003. Structure, function, and biogenesis of the cell wall of Mycobacterium tuberculosis. Tuberculosis 83, 91-97.

Gago, G., Diacovich, L., Arabolaza, A., Tsai, S.-C., Gramajo, H., 2011. Fatty acid biosynthesis in actinomycetes. FEMS Microbiology Reviews 35, 475-497.

*Figure 3: although the abundance of mycobacteria are highest near the seep, the lowest del13C values actually occur 1.8m from the seep. One might (incorrectly) assume that the lowest values would occur directly at the seep. Is there a suitable explanation for this?*

Unfortunately, it is not completely clear why we see these results. One explanation could be that it depends on the gas flux and how they adapt to this. It might be that mycobacteria can better utilize lower gas concentrations, similar to low affinity methanotrophs.

*Figure 2: I am not surprised that the C32 hopanoic acid dominates the non-seep samples – it appears to be the dominant hopanoic acid elsewhere (see Inglis 2018; https://www.sciencedirect.com/science/article/pii/S0016703718300036). However, I was*

*intrigued by the C31 and C33 hopanoic acids. There doesn't appear to be any mention of these in the text – I wondered if: 1) you obtained del13C values from these lipids, and 2) what the putative source of these lipids are?*

We assume that the $C_{31}$ and $C_{33}$ hopanoic acids might have similar sources as the $C_{32}$ hopanoic acids, as we shortly mentioned in section 3.2 (please see lines 224-229). Potential sources could be a range of bacteria like Alpha- and Gammaproteobacteria, Planctomycetes and Acidobacteria. We did obtain del$^{13}$C values for them; they showed the same depletion as the $C_{32}$ hopanoic acid with del$^{13}$C values ranging from -40 to -50 ‰. The del$^{13}$C values of these two minor hopanoic acids compared to the $C_{32}$ hopanoic acid were not shown in the manuscript since we felt that it did not add much value to the main story of the mycocerosic acids. We added Inglis et al., 2018 as a reference for the abundance of hopanoic acids in terrestrial environments (line 226).

*L39: I would also mention here (and elsewhere) that you can use: 1) specific bacteriohopanepolyols for methane oxidation (e.g. Talbot et al., 2016 OG; van Winden et al. 2012 GCA) and 2) hopanoid carbon isotopes (Inglis et al 2019 GCA, van Winden 2020 Geobiology) to probe methanotrophy in terrestrial environments.*

We added this info and references as requested in the introduction.

*L95: Sample collection = why 0.8 and 13.2m? Any methodological reasoning?*

We used 0 m, 1.8 m and 13.2 m to achieve different distances from the seep and the intensity of gas flux. At the main seep 0 m, there is no vegetation as well as at 1.8 m distance however the gas flux at 1.8 m is lower thus, we had two different samples from the unvegetated seep area. We took the control site at 13.2 m since we wanted to gain enough distance for the control site to the main seep area as it is known that around the Censo seep there is still microseepage in several meters distance (e.g. Etiope et al., 2002). Another reason for the 13.2 m distance is that the vegetation is similar to the general landscape around the Censo seep and is different from the 0 m and 1.8 m sites.

*L339: alternatively, the -33 to -37 per mil values could reflect a mixed bacterial community (e.g. heterotrophs + methanotrophs; e.g. Inglis et al., 2019 GCA).*

We added this info and reference to the manuscript (please see lines 356ff).

*L350: apart from having Shc gene, any other evidence the mycobacteria synthesise hopanoids? Any existing cultures?*

We addressed these comments together with similar ones from the Associate Editor above.